# Wireless Temperature Measurement for Curved Surfaces Based on AlN Surface Acoustic Wave Resonators

**DOI:** 10.3390/mi15050562

**Published:** 2024-04-25

**Authors:** Huali Liu, Zhixin Zhou, Liang Lou

**Affiliations:** 1School of Microelectronics, Shanghai University, Shanghai 201800, China; gmis@shu.edu.cn (H.L.); zzx1999@shu.edu.cn (Z.Z.); 2The Shanghai Industrial µTechnology Research Institute, Shanghai 201899, China

**Keywords:** temperature sensors array, flexible printed circuit boards, SAW resonators

## Abstract

In this paper, we propose a novel method for temperature measurement using surface acoustic wave (SAW) temperature sensors on curved or irregular surfaces. We integrate SAW resonators onto flexible printed circuit boards (FPCBs) to ensure better conformity of the temperature sensor with the surface of the object under test. Compared to traditional rigid PCBs, FPCBs offer greater dynamic flexibility, lighter weight, and thinner thickness, which make them an ideal choice for making SAW devices working for temperature measurements under curved surfaces. We design a temperature sensor array consisting of three devices with different operating frequencies to measure the temperature at multiple points on the surface of the object. To distinguish between different target points in the sensor array, each sensor operates at a different frequency, and the operating frequency bands do not overlap. This differentiation is achieved using Frequency Division Multiple Access (FDMA) technology. Experimental results indicate that the frequency temperature coefficients of these sensors are −30.248 ppm/°C, −30.195 ppm/°C, and −30.115 ppm/°C, respectively. In addition, the sensor array enables wireless communication via antenna and transceiver circuits. This innovation heralds enhanced adaptability and applicability for SAW temperature sensor applications.

## 1. Introduction

In recent years, surface acoustic wave (SAW) technology has witnessed continuous advancements. SAW sensors are one of the significant applications of SAW devices. They represent an innovative sensor that integrates SAW technology, thin-film techniques, and electronic technology, utilizing piezoelectric and inverse piezoelectric effects to achieve mutual conversion between electrical energy and sound waves [1]. The variations in the measured parameters can be reflected via changes in the wave velocity or frequency of SAW. Recent literature has reported various types of SAW sensors, which can be used to monitor physical or chemical parameters such as temperature, humidity [2], pressure [3], and flow [4] rate. Additionally, SAW devices can be powered by electromagnetic waves, and data can be retrieved by observing the echo signals of electromagnetic waves, thus freeing them from the dependency on power sources and cables. Therefore, they can serve as passive wireless sensors and have been widely researched and applied in fields such as aerospace, industrial production, and automotive electronics. Research on SAW-based temperature sensors and wireless retrieval has attracted significant interest.

For temperature monitoring, traditional temperature sensor systems based on thermistors, thermocouples, and infrared-sensitive components have become widespread. However, these methods still have many shortcomings when it comes to temperature monitoring on curved surfaces. The drawbacks of resistance temperature detectors (RTDs) developed based on flexible substrates mainly lie in the interference with their resistance caused by multiple factors [5,6,7]. In addition to temperature, mechanical deformation of the substrate also significantly affects them. When the substrate undergoes bending or stretching, the effective length of the RTD changes, leading to an increase in resistance. Therefore, the resistance stability of flexible RTDs faces certain challenges in curved applications. Infrared thermography technology faces various complex factors in practical applications, including emissivity, measurement distance, angle selection, atmospheric transmittance, and size effect (SSE). Especially in curved surface temperature measurement, the characteristics of coating emissivity varying with viewing angle lead to large errors in temperature measurement results [8]. To accurately measure the temperature in curved areas, it is necessary to comprehensively consider the projection relationship between the curved surface and the horizontal plane and then estimate the thermal conduction term. This process requires the use of contour curve equations, which are obtained via precise curve fitting of data points obtained by a three-coordinate measuring machine [9]. Due to the involvement of complex mathematical calculations and geometric relationships, the entire temperature measurement process appears particularly complex.

In previous studies, surface acoustic wave (SAW) temperature sensors were connected to their corresponding antennas via PCB boards [10,11,12]. Typically, FR-4 is used as the substrate for PCB, which exhibits poor thermal conductivity, high hardness, and limited environmental adaptability. FR-4 material has a certain moisture absorption property, and prolonged exposure to humid environments may cause material expansion and affect the performance and stability of the PCB. In recent years, research has explored the integration of flexible printed circuit boards (FPCBs) with surface acoustic wave (SAW) devices, where pre-designed interdigital transducer (IDT) electrodes on FPCBs are pressed onto piezoelectric substrates using mechanical fixtures [13,14]. However, this method imposes significant limitations on the width of the IDT electrodes. In the study, the width of the IDT electrodes is 50 μm, and its operating frequency only reaches 19.9 MHz, which appears inadequate in the current trend of high-frequency electronic device development. Additionally, this manufacturing method has limitations in terms of production speed and scale, and it is incompatible with CMOS technology. In contrast, the direct patterned fabrication of interdigital transducers on piezoelectric substrates is more mature, expedient, and efficient. It is worth noting that achieving precise alignment of two IDTs in experimental operations poses a technical challenge. It requires relying on rotating microchannels or manually adjusting the IDTs, which not only complicates the operation but may also introduce errors, potentially affecting the frequency response characteristics of the devices.

This paper achieves temperature measurement on the curved surface of objects by connecting surface acoustic wave (SAW) devices on rigid substrates to flexible printed circuit boards (FPCBs) via wire bonding. FPCB, made of polyimide (PI) as the substrate, possesses outstanding dielectric, mechanical, and wear-resistant properties at high temperatures. It finds widespread application in precision electronic fields such as aerospace, electronics, and electrical appliances. Compared to FR-4 PCBs, FPCBs are thinner and lighter, with double-sided FPCB thicknesses of only 0.11 mm, 0.12 mm, or 0.2 mm. Thus, FPCBs exhibit high flexibility and adaptability. The electronics industry is currently undergoing rapid development, with increasing demand for lightweight, thin, flexible, and miniaturized devices. The application scope of FPCBs is also expanding continuously, encompassing various fields such as wearable devices, automotive electronics, and medical health, with FPCBs gradually becoming an integral part of various electronic products. FPCBs exhibit excellent high-temperature resistance, capable of stable operation under conditions of up to 200 °C. By encapsulating surface acoustic wave (SAW) devices and antennas on FPCBs, temperature sensors can be adapted for use in more curved and complex application scenarios.

Surface acoustic wave (SAW) is a type of mechanical wave that propagates on the surface of a solid. The basic structure of SAW sensors is mainly divided into resonator-type and delay-line-type [15]. This paper adopts a single-port resonator-type SAW sensor. Compared with other types of SAW devices, SAW resonators have lower transmission losses, higher quality factors, and are more suitable for wireless communication. Surface acoustic wave (SAW) resonators are mainly composed of a piezoelectric layer, interdigital transducer (IDT), and reflector gratings on both sides. In the SAW temperature sensor, the IDT converts the electromagnetic waves received by the antenna into surface acoustic waves (SAWs), which propagate on the surface of the piezoelectric substrate. When the SAWs propagate to the reflector gratings on both sides, they superimpose to form standing waves and undergo resonance. Subsequently, the SAWs are reflected to the IDT by the reflector gratings, where the resonant SAWs are converted into electrical signals for output. Finally, the response is transmitted via the antenna on the sensor [16]. The resonant frequency is primarily determined by the velocity of the surface acoustic wave (SAW) and the width of the interdigital transducer (IDT) [17,18,19], expressed as follows:(1)$$f\approx \frac{\upsilon }{\lambda }$$

As temperature increases, the width of the interdigital transducer (IDT) expands due to thermal expansion, resulting in a reduction in the resonant frequency of the surface acoustic wave (SAW) resonator. Additionally, temperature variations impact the SAW velocity because the piezoelectric material is sensitive to environmental temperature changes, thus influencing the resonant frequency. Expanding on Taylor’s formula, the expression for the variation of the SAW resonant frequency with temperature can be obtained as follows:(2)$$f(T)=f(T_{0})+af^{\prime }(T_{0})(T-T_{0})+bf^{\prime \prime }{\left(T-T_{0}\right)}^{2}+\cdots$$
where *T_0_* is the reference temperature, *T* is the detected temperature, *f*(*T*_0_) is the resonant frequency corresponding to the reference temperature *T*_0_, and *a* and *b* are the first and second-order frequency–temperature coefficients, respectively. Since the second and higher-order coefficients are smaller than the first-order coefficient [20], the expression can be simplified to
(3)$$f(T)=f(T_{0})+af^{\prime }(T_{0})(T-T_{0})$$

Here, *TCF* reflects the sensitivity of the SAW temperature sensor. The resonant frequency of the SAW sensor corresponds to the temperature values one-to-one. Within a certain temperature range, the frequency of the sensor varies linearly with temperature. Leveraging this feature, accurate temperature values can be derived by capturing the sensor’s frequency, therefore realizing the function of wireless temperature measurement employing SAW sensors.

Sensitivity and TCF are crucial parameters reflecting the sensitivity of the device’s center frequency to temperature changes. *S* represents the relative sensitivity, defined as S=df/dT. As shown in Equation (3), *S* corresponds to the slope of the temperature frequency linear relationship. *TCF* denotes the normalized sensitivity. Combining Equation (2), the normalized sensitivity (*TCF*) of the resonant frequency to temperature changes in SAW sensors can be expressed as follows:(4)$$TCF=\frac{1}{f}\cdot \frac{df}{dT}=\frac{1}{f}\cdot \frac{\lambda d\upsilon -\upsilon d\lambda }{\lambda^{2}}\cdot \frac{1}{dT}=\frac{1}{\upsilon }\cdot \frac{d\upsilon }{dT}-\frac{1}{\lambda }\cdot \frac{d\lambda }{dT}$$

This paper presents a SAW temperature sensor array designed with AlN SAW resonators. FDMA (Frequency Division Multiple Access) technology is used to achieve temperature measurement at multiple target points in the environment. A transceiver circuit is constructed for signal processing, enabling wireless communication of the SAW sensor array. Additionally, FPCB is employed to connect the sensors to the antennas, which offers the advantage of temperature measurement on curved surfaces compared to previous SAW temperature sensors.

## 2. Materials and Methods

### 2.1. Multi-Sensor Design

In general, a SAW-based passive wireless sensing array consists of SAW resonators, antennas, and FPCBs. Figure 1 shows a physical image of multiple sensors. An impedance-matching circuit connects the antenna to the SAW resonator. To improve the performance of the sensor, it is necessary to design and optimize these three parts.

For the SAW resonator, this paper employs AlN as the piezoelectric material to meet the demands of high-temperature operation due to its higher acoustic wave velocity and excellent thermal stability at elevated temperatures [21]. The IDT adopts a split electrode structure featuring two pairs of electrodes per cycle [1,22] (Figure 1). To facilitate communication with multiple sensors, FDMA technology is utilized in this study to distinguish individual sensors in the array. FDMA technology divides the operational frequency bandwidth of the querying antenna into non-overlapping sub-bands, with each sensor assigned a specific sub-band for communication [23]. Consequently, the working frequencies of each sensor are unique, and their frequency bands do not overlap. This means that any frequency signal detected can correspond to a single sensor in the array.

Based on the operating principles of SAW resonators, this paper adjusts the interdigital width of SAW devices to select the sensor’s operating frequency range. Three SAW devices with interdigital widths of 1.15 μm, 1.2 μm, and 1.25 μm, respectively, are designed to construct the sensor array.

### 2.2. Antenna Design

Antennas are crucial components in passive wireless sensor systems. In this study, we selected a helical antenna owing to its simple structure, compact size, low cost, and ease of integration into various devices or systems. Figure 2a shows the basic structural parameters of the helical antenna. D represents the diameter of the helical antenna, d is the diameter of the antenna wire, S is the pitch, and L is the total length of the antenna. The number of turns (N) is also an important parameter of the helical antenna, typically adjusted to modulate the antenna’s frequency.

ANSYS Electronics Desktop 2021 R2 is an electromagnetic simulation software developed by ANSYS, providing an integrated platform for electromagnetic, circuit, and system simulation. HFSS (High-Frequency Structure Simulator) is a significant component of the ANSYS Electronics Desktop suite. It enables accurate modeling of structures like antennas, microstrip lines, and waveguides, followed by electromagnetic performance analysis and optimization. In this study, we utilized the HFSS component to model and simulate a helical antenna. The antenna possesses a diameter of 5 mm, a copper wire diameter of 0.6 mm, a pitch of 1 mm, and an overall length of 23 mm. The number of turns (N) is set as a design parametric, and then a parametric scan is performed on N to analyze the variation pattern of the antenna’s operating frequency for N. The scanning range is from 18 to 20, with a step size of 0.5. Based on the scanning results, adjustments and optimizations are made to the number of turns. Figure 2b presents the simulated results of the designed helical antenna. According to the simulation results, it can be observed that the operating frequency of the antenna increases as the number of turns decreases. To obtain antennas whose operating frequency matches that of the three sensors in the array, the number of turns of the antenna should be designed as 20, 19, and 18, corresponding to operating frequencies of 485.03 MHz, 502.76 MHz, and 520.89 MHz, respectively.

The radiation direction of the antenna is a fundamental performance parameter. For the Helical Antenna, the ratio of the helix diameter D to the wavelength λ determines the maximum radiation direction of the helical antenna. The ratio of the helix diameter D to λ used in this paper is less than 0.18, so the maximum radiation direction of the antenna is in the plane perpendicular to the helix axis [24]. In this case, the antenna is referred to as the normal mode. The E-plane and H-plane are added to the software to view the normalized radiation pattern of the helical antenna. The E-plane refers to the plane in which the maximum radiation direction of the antenna aligns with the electric field vector, while the H-plane refers to the plane in which the maximum radiation direction of the antenna aligns with the magnetic field vector [25]. The E-plane and the H-plane are mutually orthogonal. For the design model of the helical antenna placed parallel to the *xoz* plane, the E-plane is the *xoy* plane. In spherical coordinates, the *xoy* plane can be represented as −180° ≤ φ ≤ 180°, θ = 90°. The H-plane corresponds to the *xoz* plane, represented in spherical coordinates as φ = 0°, −180° ≤ θ ≤ 180°. From the radiation pattern in the E-plane (Figure 2c), it can be seen that the helical antenna is omnidirectional. From the radiation pattern in the H-plane, it can be observed that the antenna is directional. Therefore, the designed helical antenna meets the expected requirements and can be used for wireless communication in sensor systems.

### 2.3. FPCB Design

Due to the high operating frequencies of SAW devices, which all reach above MHz, the consistency of PCB impedance is crucial for signal integrity. To ensure stability and quality during signal transmission and to reduce signal reflection and distortion, impedance control is necessary for the PCB transmission lines connecting the SAW devices and the antenna. A common standard value is 50 Ω impedance. In PCBs with 50 Ω impedance, power loss, and signal distortion during transmission can be minimized. Additionally, this ensures impedance matching between the transmission lines and the connected antenna.

The impedance of PCBs is influenced by various factors, including dielectric constant, substrate thickness, trace width, trace thickness, and spacing between the ground plane and signal traces. According to the process parameters of PCB manufacturers, flexible printed circuit boards (FPCBs) use polyimide as the substrate, with a dielectric constant of 3.3. The thickness of double-sided boards can be selected from 0.11 mm, 0.12 mm, and 0.2 mm. In this study, the thinnest option of 0.11 mm was chosen to ensure better conformity of sensors to the surface of the object being measured. According to the manufacturer’s process capabilities, the trace thickness was set to 12 μm. It is common practice to cover a large area with copper on the bottom of PCBs, which can effectively reduce the impact of electromagnetic interference on circuits and enhance their anti-interference capability. Additionally, copper plating helps to reduce crosstalk between signals, thereby improving the quality and stability of signal transmission. For FPCBs, a 45-degree grid copper layout is typically adopted to enhance its bending and tensile strength, reducing the risk of damage and fracture caused by bending or external forces. The use of a copper grid can not only reduce the weight of the circuit board but also reduce the cost and improve the flexibility and portability of the FPCB to better adapt to a variety of applications. Therefore, by adjusting the width of the traces and the spacing between the ground and signal lines, the impedance of the transmission line can be effectively controlled to 50 Ω. 

The common types of impedance for transmission lines include single-ended impedance, differential impedance, coplanar single-ended impedance, and coplanar differential impedance. In this study, the signal lines are surrounded by copper planes, with equal spacing between the copper planes and the signal lines belonging to the coplanar single-ended impedance line. Polar SI9000 PCB is a widely used impedance calculation tool. The layer structure of FPCB includes the solder resist layer, trace layer, substrate, and copper layer. In this study, impedance calculations are performed using the Embedded Coplanar Waveguide with Ground 1B1A model in the software (Figure 3a).

According to the Sensitivity Analysis function of Polar SI9000 2022 V22.03 software, we calculated the variation curve of impedance values corresponding to the changes in the width of FPCB traces and the spacing between the ground plane and the signal line. According to Figure 3b, we can observe that as the spacing between the ground plane and the signal line increases, the impedance also increases, but the growth rate gradually slows down. When the spacing is larger than 0.15 mm, the impedance tends to be stable at about 50.5 Ω. Therefore, adopting a spacing of 0.15 mm in the design, even taking into account the process errors, the variation in impedance values is not significant. When the spacing D1 is determined to be 0.15 mm, Figure 3b reflects that the impedance is inversely proportional to the linewidth, with the larger the linewidth, the smaller the impedance value. Considering the correlation between the line width and the difficulty of the process, we finally designed the line width as 0.8 mm, with a tolerance of ±20% for the line width. According to Figure 4b, the variation range of impedance within the line width tolerance is from 46 Ω to 56 Ω, which complies with the industry-standard error criterion of ±10%.

### 2.4. Wireless Reading System Design

The experimental setup for wireless temperature sensing mainly consists of sensors, interrogation antennas, transceiver circuits, and oscilloscopes. A commercially available antenna with high gain and a working bandwidth of 470 MHz–530 MHz is used for reading the circuit. Figure 4 shows the system diagram of the wireless reading circuit [16,26,27,28]. At present, the reading circuit of wireless passive resonant sensors is mainly divided into time–domain sampling and frequency–domain sampling. The basic principle of time–domain sampling is to generate pulse signals using a frequency source, transmit them to the outside via the antenna to excite the sensor, and then detect and analyze the feedback echo signal; the resonant frequency information of the sensor is obtained. The time–domain sampling method can read the signal quickly and satisfy the need for fast signal acquisition, so this research adopts the time–domain sampling method. Then, we use an oscilloscope to convert the time–domain signal into the frequency–domain signal and read the frequency value of the echo signal. By the correspondence between temperature and frequency, we can calculate the temperature value of the object to be measured.

The reading circuit controls the receiving and sending state of the circuit via the RF switch and realizes the isolation of the different path signals to prevent the interference between the transmitting signal and the receiving signal. The FPGA generates square wave signals within a certain period. When the level is high, the RF switch is selected to connect the antenna with the transmitting circuit. At this time, the PLL generates an inquiry signal corresponding to the resonant frequencies of the three sensors. After amplification by the RF power amplifier, the signals are transmitted by the antenna and received by the SAW sensors. When the Square wave is low, the RF switch connects to the other end of the circuit, putting the reading circuit in a receiving state. At this time, the SAW sensor returns the echo signal with temperature information. After the antenna receives the signal, it passes through an amplifier and filter. Then, the return signal *f_1_* and the polling signal *f*_0_ generated by the PLL are input to the mixer then selects the output IF sign via the band-pass filter [29]. The IF signal is connected to the oscilloscope (KEYSIGHT, Santa Rosa, CA, USA); then, the FFT function is used to convert the time domain signal into the frequency domain signal, and the frequency of the if signal Δf is obtained. Figure 5 illustrates the passive wireless temperature measurement experimental platform we constructed. 

## 3. Experiments and Results

### 3.1. Simulation Methodology

In this study, a two-dimensional model of the SAW device is established by using the Finite Element Simulation Software COMSOL Multiphysics 6.0 to calculate the S-parameters and resonant modes of each sensor. SAW devices usually consist of hundreds of identical periodic electrodes, where the length of each electrode is much greater than its width [30]. Creating a 3D model for all periods would result in complex models and longer computation times. Because of the periodicity of the IDT structure, it is feasible to simulate infinitely long-period SAW devices in COMSOL by adding periodic conditions on both sides of a single-period 2D model. an infinite period SAW device can be simulated in COMSOL by adding periodic conditions to the left and right sides of the single-period 2D model [31]. This approach not only ensures the accuracy of the simulation but also significantly improves the computation speed and efficiency. Figure 6a shows the established two-dimensional simulation model. The top layer consists of a Mo electrode with a thickness of 0.2 μm; the piezoelectric layer consists of an AlN film with a thickness of 1 μm, and the substrate is silicon. The IDT of three kinds of sensors all adopt uniform transducer, and the ratio of interdigital width to interdigital distance is 0.5. Therefore, the relationship between finger width and wavelength can be expressed as a = λ⁄8. Considering that the energy of the surface acoustic wave is mainly concentrated within a range of 1–2 wavelengths below the solid surface, we simplify the substrate thickness to be equal to 2λ.

In the simulation of the model, the propagation modes of SAW are observed via the analysis of characteristic frequencies. Figure 6b,c illustrates two different vibration modes of Rayleigh waves, where Figure 6b represents anti-resonance and Figure 6c represents resonance. According to the simulation results, the energy in the Rayleigh Wave is mainly concentrated in the 1–2 wavelength range below the solid surface, so it is highly sensitive to temperature changes [2,32]. Furthermore, the frequency range and step length are added to calculate the frequency response of the model under excitation, and the reflection parameter S_11_ is obtained. Figure 7 shows the S-parameter curves of three SAW devices with different interdigital widths, where the interdigital widths are 1.15 μm, 1.2 μm, and 1.25 μm, and the corresponding central frequencies are 522.902 MHz, 502.639 MHz, and 483.918 MHz, respectively.

### 3.2. SAW Resonator Testing in Terms of Temperature

The reflection parameters S_11_ of three SAW devices were measured by vector network analyzer at room temperature and compared with the simulation results. The prefabricated SAW devices were pasted onto the designed FPCB using thermally conductive silicone adhesive and left for 8 h to ensure the stability of the SAW. Then, we use wire bonding technology to connect the SAW resonator to the FPCB pad by thin metal, realizing signal transmission between the chip and FPCB. SMA connectors are used as interfaces between SAW devices and test instruments. By analyzing the S-parameter curve, we can determine the operating frequency and evaluate the performance of the device. The reflection parameter S_11_ of three SAW devices was measured using a VNA (Rohde&Schwarz ZNL6, Rohde&Schwarz, Munich, Germany). The instrument needs to be calibrated before performing the measurements to ensure the accuracy of the results. At room temperature, the measured operating frequencies of the three SAW devices are 484.144 MHz, 503.733 MHz, and 525.739 MHz, which are consistent with the simulation results. However, due to the different material preparation processes, there were certain discrepancies between the physical parameters of the substrate, piezoelectric thin film, and electrodes used in the study and those set in the simulation, leading to incomplete alignment between the simulated and experimental results. Additionally, factors such as the uniformity of the finger electrode edges during the photolithography process, the occurrence of polycrystalline phenomena during electrode sputtering, and the interface continuity between the electrode and the piezoelectric material differed from the ideal conditions set in the simulation [25,33].

After testing the reflection parameter S_11_ of the SAW resonator with three different finger widths at room temperature, we placed the SAW resonators on a heating table and utilized a VNA to measure the trend of their S-parameter curves with temperature variations. The experimental temperature range was from room temperature to 100 °C. Figure 8 illustrates the offset results of the S_11_ curves of the three SAW resonators with temperature variations. It can be observed that the S_11_ curves are shifted to the left, and the operating frequency decreases as the temperature increases. There is a clear negative correlation between frequency and temperature. We conducted two heating–cooling experiments separately for each of the three SAW sensors. At each temperature point, four frequency values were obtained, and their average was calculated as the operating frequency at that temperature. Additionally, we calculated the standard deviation and set it as the error bars. According to Figure 8b,d,f results, it can be observed that the operating states of the three devices were relatively stable, and multiple experimental results did not show significant fluctuations.

After linear fitting of the experimental results, the obtained outcomes are shown in Figure 8b,d,f. Therefore, it can be concluded that all three SAW devices exhibit good sensitivity and linearity, making them suitable for use as temperature sensors. According to the expressions for sensitivity and TCF (Equation (4)), we calculated the sensitivity and TCF of each temperature sensor. For the SAW device with a finger width of 1.15 μm, the sensitivity is −15.91 kHz/°C, and the TCF is −30.248 ppm/°C; for the SAW device with a finger width of 1.2 μm, the sensitivity is −15.21 kHz/°C, and the TCF is −30.195 ppm/°C; for the SAW device with a finger width of 1.25 μm, the sensitivity is −14.58 kHz/°C, and the TCF is −30.115 ppm/°C. Table 1 summarizes the performance characterization results of the three saw sensors. Table 2 compares the TCF of temperature sensors based on SAW devices. It is evident that the TCF results reported in this paper are acceptable compared to previous studies.

Based on the experimental results, we observe some regularities: for SAW resonators with different finger widths, narrower finger widths correspond to higher sensitivity S. Although as the finger width decreases, the TCF also increases, the growth rate is relatively slow. This indicates that SAW resonators with higher operating frequencies can effectively enhance sensor sensitivity, but TCF is influenced to a certain extent by the parameters of the piezoelectric material [19].

### 3.3. Wireless Passive SAW Temperature Sensors Array Measurement

According to the wireless passive experimental test platform shown in Figure 5, the temperature sensor array was measured wirelessly. Three sensors connected to respective antennas were placed on the surface of the curved heating table and thermally bonded using thermally conductive double-sided adhesive. At the same time, a thermocouple is placed near the SAW sensors to calibrate the temperature. The distance between the antenna and the heating platform is 20 cm. The response time of the device is about 0.15 μs. With the increase in temperature, the frequency of the echo signal decreases gradually while the reading signal remains unchanged, so the ∆*f* of the intermediate frequency signal increases gradually. Therefore, the rate of change of if signal is opposite to the sensitivity of the sensor. Figure 9 shows the results of the wireless measurement experiment with a set of sensor arrays. By analyzing the rate of change of the signal, the sensitivity and linearity of the three sensors in wireless measurement were evaluated and compared with the results obtained by Vector Network Analyzer (VNA) measurements. According to the fitting results, all three sensors show good linearity with the sensitivity of −14.604 kHz/°C, −15.418 kHz/°C, −16.251 kHz/°C and temperature coefficient TCF of −30.323 ppm/°C, −31.184 ppm/°C, −31.610 ppm/°C. These results are basically consistent with the experimental results obtained from wired testing. 

## 4. Discussion

With the development of SAW temperature sensors, there is a growing demand for their application in fields such as aviation and automotive electronics. In this study, a SAW temperature sensor array based on FPCB is proposed to address the complexity of current application environments. The sensor array uses FDMA technology to encode and realize multi-point temperature measurements. Leveraging the flexibility and thinness of FPCBs, the flexible SAW sensor can fit the curved object better. A two-dimensional model was established using the finite element simulation software COMSOL to calculate the operating frequencies of SAWs with different finger widths. The simulation results are verified by VNA, and the temperature–frequency characteristics of the sensor are characterized. The results show that the three devices have good linearity and sensitivity, which proves the feasibility of the flexible SAW temperature sensor array. Then, a reading circuit system for the sensors is further designed. A group of SAW sensors with three different working frequencies communicate wirelessly via a single inquiry antenna, which verifies the wireless temperature measurement capability of the sensor array. 

In the subsequent work, further simulation and testing can be conducted to characterize the strain performance of the AlN surface acoustic wave resonators used in this study in order to assess the flexibility of the devices themselves and to delve into the reliability and stability of the FPCB, AlN piezoelectric layer, and interdigital transducer (IDT) under repeated bending. Moreover, alternative solutions to the spiral antenna can be explored, considering the characteristics of antenna miniaturization, high gain, and frequency adjustability, to further optimize and enhance the ability of the surface acoustic wave temperature sensor to monitor multi-point temperature information in environments with wiring difficulties and uneven surfaces.

## Figures and Tables

**Figure 1 micromachines-15-00562-f001:**
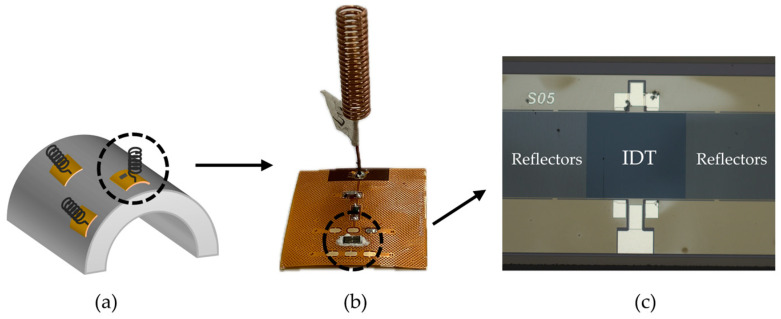
(**a**) Three temperature sensors conforming to the surface of a curved object. (**b**) Physical image of a single temperature sensor. (**c**) Microscope image of the interdigital transducer structure.

**Figure 2 micromachines-15-00562-f002:**
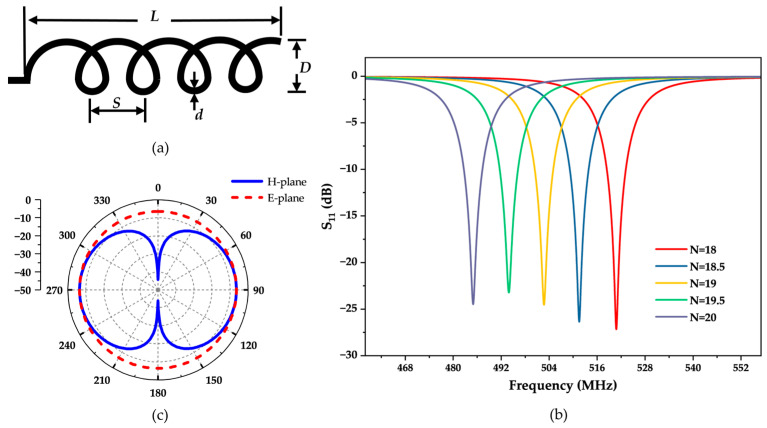
(**a**) The basic structural parameters of the helical antenna (**b**) Simulated reflection profile (S_11_) of the designed helical antenna varying with the turn number N. (**c**) The normalized radiation pattern of the antenna.

**Figure 3 micromachines-15-00562-f003:**
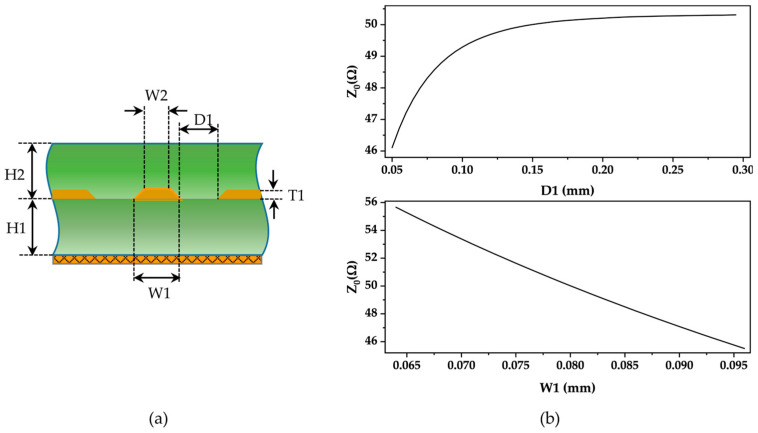
(**a**) Embedded Coplanar Waveguide with Ground 1B1A. (**b**) The variation of impedance (Z0) with the spacing between the ground line and the signal line (D1) and the linewidth (W1).

**Figure 4 micromachines-15-00562-f004:**
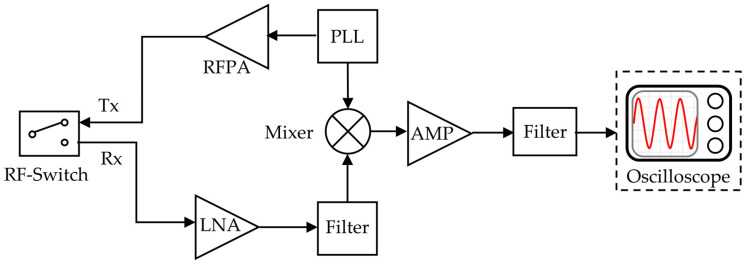
A simple block diagram of an implemented transceiver.

**Figure 5 micromachines-15-00562-f005:**
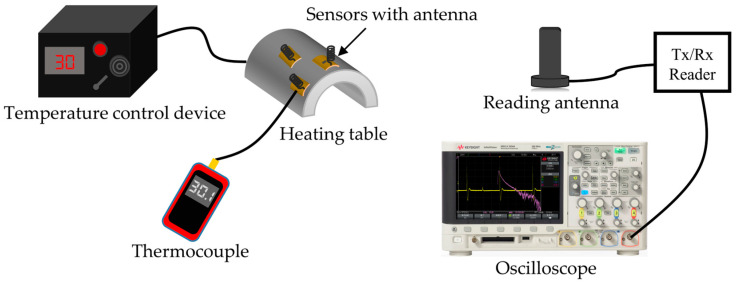
Experimental set-up for the wireless measurement on the curved heating table.

**Figure 6 micromachines-15-00562-f006:**
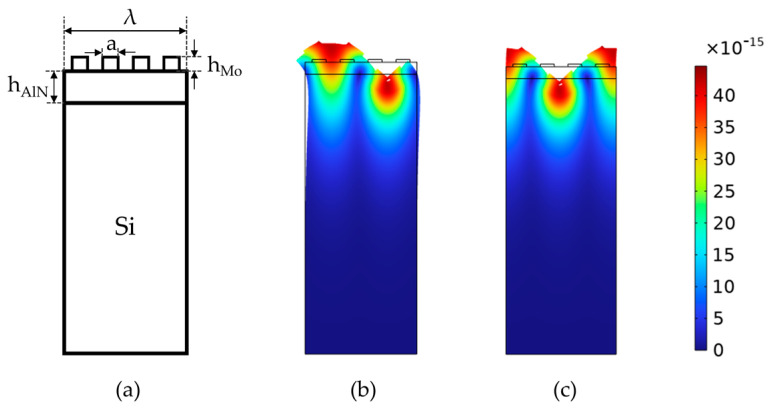
Simulation component. (**a**) Two-dimensional periodic model of the Mo/AlN/Si structure. (**b**) The antisymmetric modes of the Rayleigh wave. (**c**) The symmetric mode of the Rayleigh wave.

**Figure 7 micromachines-15-00562-f007:**
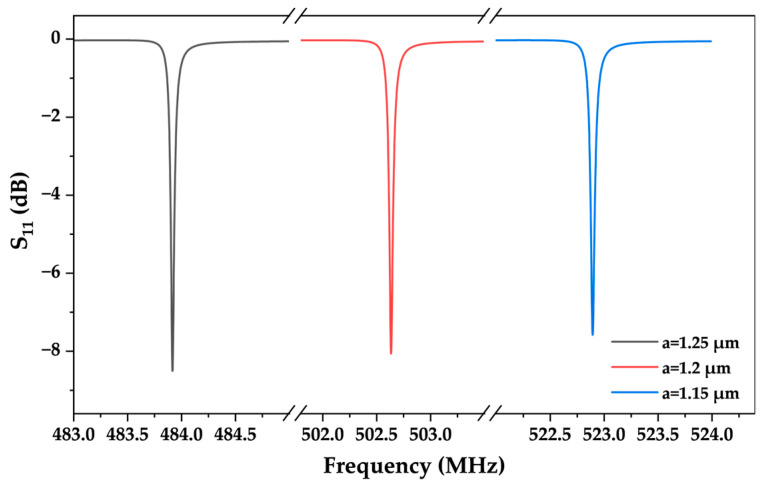
Simulation result of reflection parameter(S_11_) for three finger widths (1.15 μm, 1.2 μm, and 1.25 μm).

**Figure 8 micromachines-15-00562-f008:**
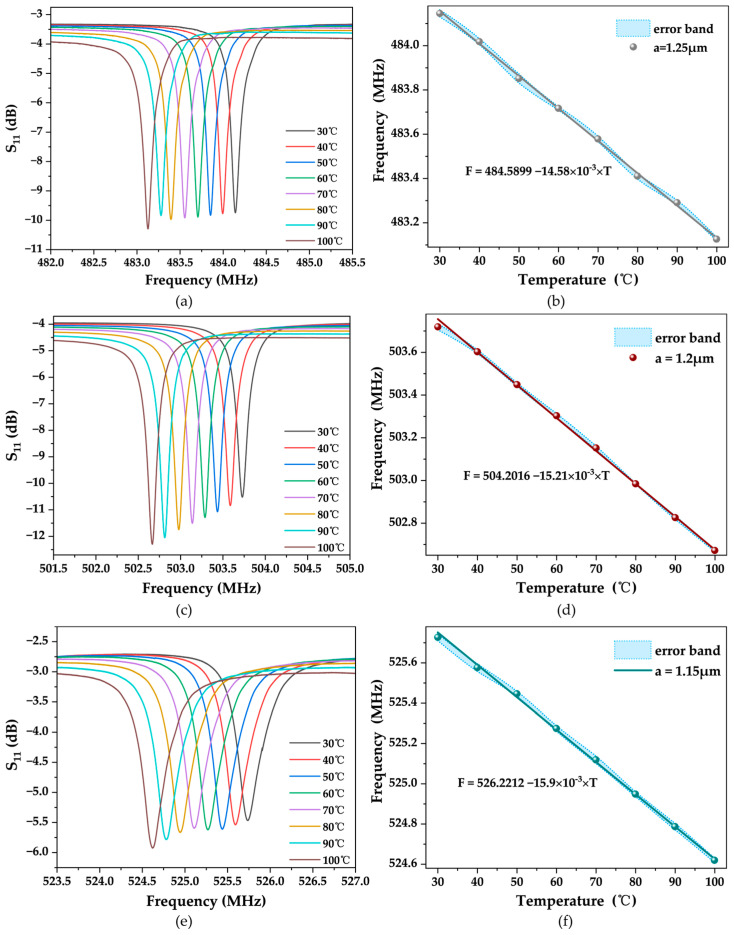
Temperature characterization of sensors with different IDT widths. (**a**,**c**,**e**) are the S_11_ curves for IDT widths of 1.25 μm, 1.2 μm, and 1.15 μm at different temperatures. (**b**,**d**,**f**) are the resonant frequency versus temperature for IDT widths of 1.25 μm, 1.2 μm, and 1.15 μm.

**Figure 9 micromachines-15-00562-f009:**
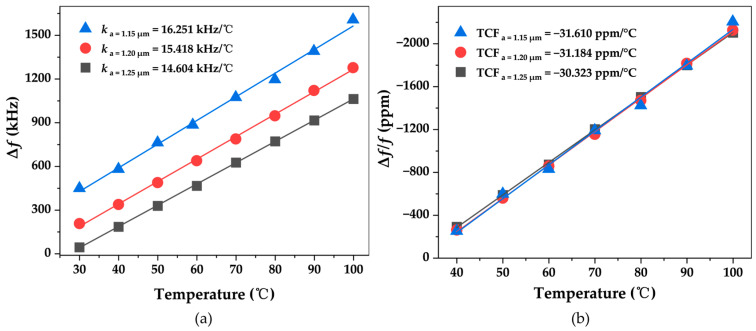
Wireless measurement results. (**a**) Frequency variation of IF signals with temperature; (**b**) three sensors of relative frequency versus temperature.

**Table 1 micromachines-15-00562-t001:** Performance measurements of the SAW sensors array.

Parameter	Device 1	Device 2	Device 3
IDT electrode width (μm)	1.15	1.2	1.25
Center frequency (MHz)	525.727	503.719	484.146
Sensitivity (kHz/°C)	−15.91	−15.21	−14.58
TCF (ppm/°C)	−30.248	−30.195	−30.115

**Table 2 micromachines-15-00562-t002:** Comparison of the TCF between the device we studied and previously reported sensors.

Ref.	Structure	TCF (ppm/°C)
1 [34]	LN/SiO_2_/Si	−26.01
2 [35]	AlN/Pt/LGS	−14.55
3 [36]	Mo/AlN/Mo/Si	−27.2
4 [37]	Mo/AlN/SiO_2_	−24.52
5 [38]	Mo/AlN/Si	−30

## Data Availability

Data are contained within the article.

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
