# Peer review of "Wireless Temperature Measurement for Curved Surfaces Based on AlN Surface Acoustic Wave Resonators"

_micromachines, 2024, doi:10.3390/mi15050562_

Round 1
Reviewer 1 Report
Comments and Suggestions for Authors
The authors demonstrated a flexible temperature sensor array utilizing Surface Acoustic Wave (SAW) technology and carried out temperature experiments. The experimental results proved the feasibility of the proposed sensors. However, I am afraid this manuscript may not be published on the journal at current state.
It seems that the novelty failed to be presented. Research on Wireless SAW temperature based on AlN film has been intensively reported. The flexible scheme seems nothing to do with SAW, instead, it is subject to the PCBs. I may remind the authors of the scope of the special issue and journal. At least, enough content should be related with acoustic micromachines. Overall, this manuscript depicts more like a technical report than a scientific paper.
Comments on the Quality of English LanguageEnglish language fine. No issues detected.
Author Response
Thank you for your valuable feedback regarding my manuscript. My reply is in the uploaded document.

Reviewer 2 Report
Comments and Suggestions for Authors
This communication report detailed the design and performance of a wireless temperature sensor based on an AlN SAW resonator on a flexible PCB. The description and experiment were written with due diligence but some experiments are missing for the author's certain claims. I have detailed the major points below, and the paper can be reconsidered for publication if addressed properly.
(1) Define "flexible" with quantitative experiments - one major claim the authors are proposing is their implementation of an AlN SAW resonator on flexible PCB. Below are the questions -
1-1. If I understand correctly, one can do the exact same implementation of the AlN SAW resonator on a rigid substrate and achieve the same performance. In the manuscript, the authors mentioned about the advantage of the PI substrate that a flexible PCB uses which I would agree, however, if that was the only thing, then this paper seems to present a device that was built on a flexible PCB, which I don't see any novelty, as many similar devices have been built on FPCB. I don't find the special/new things about this AlN SAW resonator, please justify the innovation here. I also cannot find the fabrication methods of AlN saw onto this PCB. Please elaborate in your report.
1-2 If it is "flexible", how "flexible" it actually is? The missing experiment is a bending test to show the limit of this device, how flexible they can be, and an associated reliability test should be performed at the same time to show that flexibility is durable. I want to note that the antenna is relatively bulky and it only has a welding point to the substrate which is very prone to break. AlN is not supposed to be a flexible material, does it limit the flexibility of the device? How?
Please show some data first and also comment on any alternative to replace this form of antenna, otherwise, it is doubtful it can be a robust device.
Figure 1 also requires detailed labeling on each component in each image. Please fix it.
(2) The performance of the said device lacks benchmarking.
2-1 Please elaborate your device performance to prior art, ie, temperature sensor based on SAW or other mechanisms on rigid or flexible substrates, how is your sensitivity/linearity compared to them?
2-2 What is the response time of your device? Please elaborate. Also, please include error bars to your data in Figure 9 and 10 for data fidelity.
2-3 What is the noise level for your device? Does it have temperature dependence?
Comments on the Quality of English LanguageMany descriptions are too wordy and can be simplified. Please find native speaker to polish if possible.
Author Response

(The authors gave the same response as above.)

Round 2
Reviewer 1 Report
Comments and Suggestions for Authors
Authors have revised the manuscript. No further comments
Comments on the Quality of English LanguageAuthors have revised the manuscript. No further comments
Author Response
Thank you for your valuable feedback regarding our manuscript.
Reviewer 2 Report
Comments and Suggestions for Authors
Authors have made significant efforts in addressing my comments. However, some are still unsatifactory. See my detailed responses to authors' responses and corresponding action items to take.
Response 1-1: thanks for the due diligence of the extensive literature review to respond this quest. Please make sure all these additional comments are incorporated into the main text with concise formats. The new data can be incorporated into SI.
Response 1-2:
1 - The information regarding the FPCB flexibility is appreciated. Please incorporate these information into main text or SI for completeness of the research. However, there are two more information are needed - (1) does the curvature limit of FPCB applies to AlN thin film? What is the "flexibility" of AlN? Some literature search is needed. (2) how durable is the FPCB and AlN and also the metal electrodes are under repeated bending? Please provide some information that is available in the literature.
2- the effort on ceramic version of the antenna is appreciated. Please incorporate this information in the main text (as texts) with the figures in SI for completeness of the research. It would also be beneficial to check some mechanical methods to make sure your antenna can be robustly fixed on the FPCB based package without breaking.
I understand actual bending test might be hard to do, but I suggest you can comment such as a to-do list as this is important eventually if such devices can be implemented in the field.
Response 2-1 - a figure of merit comparison table is important. Please incorporate such in your main text.
Response 2-2 - please make sure the time response data is mentioned in your main text. For the error bars, you need to include that in your main text figures no matter what. It make no sense to exclude that due to they are small values compared to the scale, and that is not able to be seen by human eyes. It is the most basic scientific practice and everyone should follow.
Response 2-3 - understood. But, you still need to comment that the noise level is lower than the resolution of the oscilloscope measurable level with the specification (ie, what is the oscilloscope resolution?). Please be as rigorous as possible when publishing your results.
Author Response

(The authors gave the same response as above.)
